# ATP-Dependent Chromatin Remodellers in Inner Ear Development

**DOI:** 10.3390/cells12040532

**Published:** 2023-02-07

**Authors:** Ilyas Chohra, Keshi Chung, Subhajit Giri, Brigitte Malgrange

**Affiliations:** Developmental Neurobiology Unit, GIGA-Stem Cells, Av. Hippocrate 15 B, 4000 Liege, Belgium

**Keywords:** cochlea, hair cells, epigenetic, differentiation, development

## Abstract

During transcription, DNA replication and repair, chromatin structure is constantly modified to reveal specific genetic regions and allow access to DNA-interacting enzymes. ATP-dependent chromatin remodelling complexes use the energy of ATP hydrolysis to modify chromatin architecture by repositioning and rearranging nucleosomes. These complexes are defined by a conserved SNF2-like, catalytic ATPase subunit and are divided into four families: CHD, SWI/SNF, ISWI and INO80. ATP-dependent chromatin remodellers are crucial in regulating development and stem cell biology in numerous organs, including the inner ear. In addition, mutations in genes coding for proteins that are part of chromatin remodellers have been implicated in numerous cases of neurosensory deafness. In this review, we describe the composition, structure and functional activity of these complexes and discuss how they contribute to hearing and neurosensory deafness.

## 1. Introduction

Eukaryotic chromatin is densely compacted in the nucleus through nucleosomes, an assembly of DNA wrapped around an octamer of two copies of the four histone proteins H2A, H2B, H3 and H4 [1]. Chromatin accessibility is the level to which DNA-interacting proteins can physically contact DNA regulatory elements and it is determined by the occupancy and topological architecture of chromatin [2]. During development, cells commit to strictly controlled programs that ensure each tissue and organ’s proper specialisation and function. The epigenetic profile of each cell defines these programs, and their transition is made possible by the elasticity of chromatin, which is safeguarded by several factors including, amongst others: histone chaperones, nucleosome post-translation modifications and chromatin remodellers [3,4].

Inner ear development is a heavily controlled process governed by dynamic epigenetic regulation, consisting of several mechanisms that ensure the spatiotemporal control of gene expression—one of the fundamental mechanisms to ensure chromatin remodelling. This ATP-dependent mechanism is carried out by a highly evolutionarily conserved set of enzymes belonging to the SNF2 superfamily called chromatin remodellers which are involved in many cellular processes such as transcription, recombination, DNA repair and replication [5]. They are categorised into four families according to their protein similarity and domain structure: CHD (chromodomain-helicase-DNA binding), SWI/SNF (switch/sucrose-non-fermenting), ISWI (imitation switch) and INO80 (inositol-requiring 80) (Table 1).

The inner ear is a multifunctional sensory organ responsible for hearing (cochlea), balance and orientation in space (vestibular system). Following a thickening on the ectoderm surface, the pre-placodal region will give rise to the otic placode at the end of the third week of human gestation. A series of signalling molecules such as FGFs highly regulate this process. The otic placode then invaginates to form the otocyst that goes on to morph into different structures of the inner ear through the modulation of gradients of morphogenic factors. An opposing gradient of Wnt and sonic hedgehog (Shh) defines the dorsoventral axis, a gradient of retinoic acid (RA) defines the anteroposterior axis and at least in part, FGF3 regulates the specification of the mediolateral axis [6]. The adult auditory portion of the inner ear (i.e., the cochlea) is composed of several cell types, that include sensory hair cells (HCs) of the organ of Corti, which are organised in one row of inner hair cells (IHCs) and three rows of outer hair cells (OHCs), surrounded by supporting cells (SCs) that play a crucial role in the development, function and maintenance of the HCs [7]. The HCs are characterised by their apical stereocilia bundle, disrupted in many hearing defects.

## 2. CHD Family

The CHD family consists of nine members (CHD1–CHD9). Their structure is characterised by two consecutive chromodomains in the N-terminal region and an SNF2-like ATP-dependent helicase domain positioned in the central region. They recognise and bind nucleosomes to contribute, in many cases, to the formation of heterochromatin typically marked by the presence of methylated histones and other repressive chromatin markers. CHD proteins can bind to methylated histones through their chromodomains and use their helicase activity to remodel the chromatin and contribute to the formation of heterochromatin, a fundamental feature of chromosomes that ensures genomic stability [8].

CHD proteins are classified into three subfamilies (Class I–III) (Figure 1), defined by significant structural motifs and their association with specific complexes. Class I comprises CHD1 and CHD2, characterised by an additional C-terminal DNA-binding domain with a preference for binding AT-rich DNA sequences [9]. CHD3, CHD4 and CHD5 are class II CHD proteins, identified by a pair of N-terminal plant homeodomain (PHD) zinc finger domains and a lack of a DNA-binding domain [10]. In addition, these three paralogues present two C-terminal domains of unknown function (DUFs), DUF1087 and DUF1086 [11]. CHD6, CHD7, CHD8 and CHD9 are class III CHD proteins unique to metazoans. They often include a C-terminal duplicated Brahma and Kismet (BRK) domain classically found in proteins involved in transcription and developmental signalling in higher eukaryotes [12].

Regardless of overall protein domain structure, the function of CHD superfamily proteins is directly connected to regulating gene expression by controlling chromatin states. They are mainly translational repressors [13]. Here, we describe only the CHDs showing evidence of implication in the inner ear.

**Figure 1 cells-12-00532-f001:**
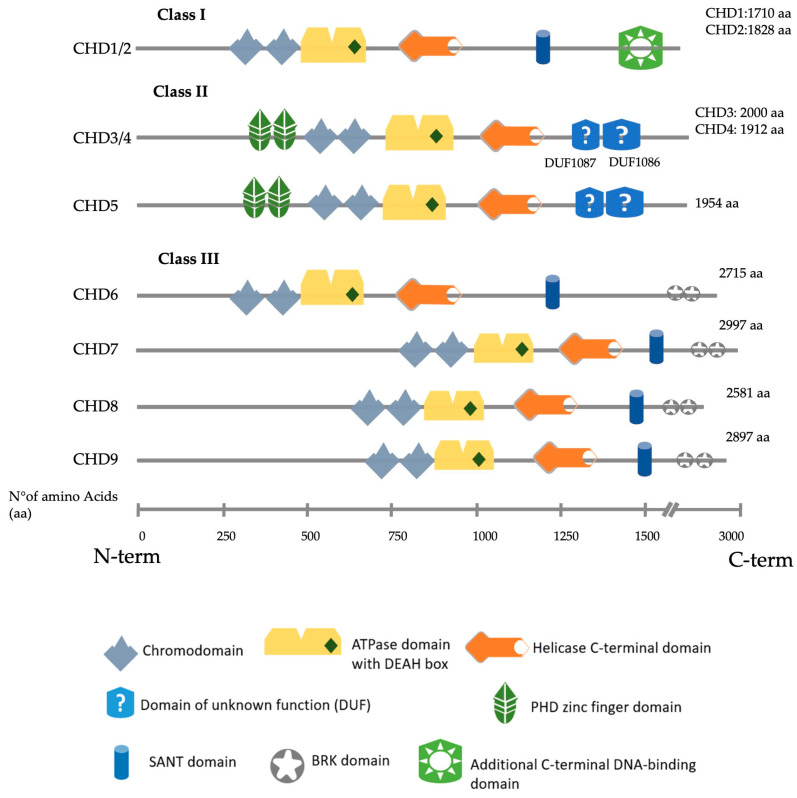
Schematic representation of all nine human CHD proteins divided by their class, showing the approximate position of the main predicted domains, as described in Uniprot, SMART and InterPro [14,15,16].

### 2.1. CHD3/4

CHD3 and CDH4 share 71.6% of their amino acid sequences. Both proteins form distinct nucleosome remodelling deacetylase (NuRD) complexes with different but overlapping functionalities [17]. Mass spectrometry data show that CHD3 and CHD4 may form heterodimers within NuRD complexes [18]. CHD3 and CHD4 are the core subunit of the NuRD complex [19]. Their binding to this complex is mutually exclusive [20]. Normal NuRD complexes are essential for many developmental and cell differentiation processes, including maintaining embryonic stem (ES) cell pluripotency and regulating progenitor cell development in numerous organs such as the brain, hematopoietic system or kidney [21,22].

Snijders Blok–Campeau syndrome is a recently described syndrome that encompasses patients with variants in *CHD3*, characterised by intellectual disability, developmental delay (especially speech delay) and several dysmorphic features [23], with most of the mutations localising within the ATPase or the helicase domain [24]. A minority of these patients manifest neurosensory hearing loss (3 out of 24). More investigations are needed to understand further the implications of hearing deficit in this syndrome.

Mutations in the CHD4 protein result in Sifrim–Hitz–Weiss (SIHIWES) syndrome, an autosomal dominant neurodevelopmental disease. Symptoms identified include global developmental delay, mild to moderate intellectual disability, brain anomalies, congenital heart defects, dysmorphic features, macrocephaly and conductive and/or sensorineural hearing loss in 55% of the cases [25]. The mutations identified in hearing loss patients affect the ATPase domain, the PHD domain or the DUF1087 domain (Figure 2) [25,26].

In mice, Chd4 is present in the organ of Corti, starting at embryonic day 18.5 (E18.5) throughout most cells. It remains present during postnatal development, and its expression is restricted to SCs at the adult stage (postnatal P21) [27]. In the central nervous system, Chd4 is strongly expressed in both neural progenitors and mature neurons. Conditional knockout of Chd4 in nestin-positive neural progenitors in the mouse cerebral cortex leads to their precocious cell cycle exit and apoptosis, thereby causing microcephaly [28]. It has been suggested that Chd4 regulates neural differentiation by controlling the cell cycle through repressing the acetylation of p53 and possible transcription activation of neural differentiation genes [22]. Whether Chd4 has the same effect in the cochlea remains to be established.

Altogether, these findings on *CHD4* mutations in hearing loss patients and the *Chd4* pattern of expression in embryonic mice cochlea suggest that CHD4 may influence spatiotemporal gene expression regulation in the developing cochlea. This deficiency could lead to deafness. Further experimental evaluation is critical to decode the mechanistic role of CHD4 in otic development and how particular missense mutations can give rise to the deafness phenotype.

### 2.2. CHD7

De novo heterozygous mutations in the *CHD7* gene are the leading causes of CHARGE syndrome, a complex neurodevelopmental disorder characterised by ocular Coloboma, Heart defects, Atresia of the choanae, Retarded growth and development, Genital hypoplasia and Ear abnormalities, including mixed conductive/sensorineural hearing loss [29,30]. More than 500 heterozygous *CHD7* mutations have been identified in CHARGE syndrome patients, most of which are nonsense, frameshift indels, splice site and missense mutations [31].

In mice, Chd7 is highly expressed throughout the developing otocyst starting at E9.5. Later during development, Chd7 is present in the cochlear epithelium both in HCs and SCs and in spiral ganglion neurons. Chd7 is still present postnatally but is decreased in the organ of Corti [32,33].

A Chd7 heterozygous mouse model was generated from Chd7-deficient, gene-trapped lacZ reporter ES cells (hereafter Chd7^Gt/+^ mice) that expresses β-galactosidase under the control of the Chd7 promoter [34]. Homozygous Chd7^Gt/Gt^ are embryonic lethal. HC formation and cochlear innervation were normal in Chd7^Gt/+^ mice. However, Chd7^Gt/+^ animals present mild hearing loss with elevated auditory brainstem recordings (ABR) and reduced distortion product otoacoustic emissions (DPOAE) [34]. Recently, a re-evaluation of Chd7^Gt/+^ suggests a new function in the development and maintenance of satellite glial cells in the spiral ganglia, as well as in the regulation of myelin sheaths surrounding type I spiral ganglion neurons that innervate the sensory epithelium of the cochlea [35]. In two ENU-induced mutations in Chd7—looper and trooper strains—having a nonsense mutation (c.5690C>A, p.S1897X) or a cryptic splice site mutation (c.3219-18T>A), respectively, middle ear and vestibular defects were found to be a prominent phenotype with early onset hearing loss [36,37].

Deficiency in Chd7 during development has also been reported to specifically impact neuronal development in the inner ear. In a Foxg1-Cre-driven conditional Chd7 knockout mouse, the otocyst had a reduced vestibulocochlear ganglion size. This smaller neuronal number is due to the reduced expression of the neuronal fate-specific genes Ngn1, Otx2 and Fgf10, leading to reduced neuronal cell proliferation [32,38]. A recent study has shown that Chd7 deficiency leads to IHC and auditory neurons being vulnerable to ototoxic stress, rapid postnatal degeneration and profound hearing loss [39].

Modulating retinoic acid signalling prevents inner ear defects caused by Chd7 deficiency [40]. Indeed, loss of the retinoic acid (RA) synthetic enzyme Aldh1a3 or treatment with citral—an inhibitor of RA synthesis—partially rescued the semicircular canal deformation phenotype in Chd7^Gt/+^ mice [41]. Upregulation or downregulation of RA signalling during embryogenesis leads to developmental defects similar to those observed in CHARGE. RNA-seq and qRT-PCR combined with ChIP experiments in inner ears demonstrate that Chd7 acts upstream of *Aldh1a3* via direct binding and repression of *Aldh1a3* [41].

To evaluate the Chd7 deficiency at single-cell resolution, Durruthy-Durruthy et al. performed single-cell multiplexed qPCR in Pax2Cre; mT/mGFP otocysts cells from Chd7+/+ and Chd7^Gt/+^ mice for 192 inner ear specific genes and reported a cellular identity shift towards neuroblasts in E10.5 otocysts [42]. This cell fate decision may arise through disruption in the pro-sensory and pro-neurogenic gene expression network and Notch signalling pathways.

To model the inner ear phenotypes in CHARGE syndrome in a human context, Nie et al. adopted a previously published human induced pluripotent stem (hiPS) cell/human ES cell-derived otic organoid model [43]. The study reported a complete loss of inner ear HC formation in the CHD7 knock-out line (*CHD7*^−/−^) or CHD7 patient-specific missense mutation (CHD7^S834F/S834F^) d70 otic organoids [44]. In contrast, the haploinsufficient/heterozygous otic organoids (CHD7^S834F/+^ or CHD7^+/−^) could generate HCs. To understand the molecular pathogenesis, the CHD7 mutations were created in a WA25 PAX2nGFP hESC line, which allows for the study of the transcriptome of *PAX2*^+^ otic progenitors from both CHD7^+/+^ and CHD7^−/−^ d20 organoids. Single-cell RNA-sequencing (scRNA-seq) transcriptome data on those PAX2+ cells revealed a downregulation of several essential inner ear morphogenesis and developmental genes such as *TBX1*, *LMX1A*, *SOX10*, *DLX5* and *SIX1* and disruption in FGF, BMP, Notch, TGF-β and Wnt signalling pathways.

These effects on PAX2 otic progenitors were reflected in HC development. Indeed, scRNA-seq on POU4F3-nd tomato hair cells from *CHD7*^+/−^ d70 organoids showed numerous dysregulated deafness genes compared with the WT counterparts. *SIX1*, *USH1C* and *STRC* were downregulated, giving potential explanations for hearing loss observed in patients with CHARGE syndrome. Co-differentiating both WT and *CHD7*^−/−^ hESCs in a chimeric organoid system partially restored the expression of otherwise severely downregulated essential otic genes *FBXO2*, *SOX10* and *DLX5* but failed to generate any hair cells in the CHD7^−/−^ background [44]. Altogether, this study described the critical role of CHD7 in otic lineage differentiation and hair cell development and, more importantly, in an *in-vitro* humanised experimental model.

### 2.3. CHD8

CHD8 is most prominently known for its association with autism spectrum disorders (ASD) [45]. CHD8 has two isoforms: CHD8L, a full-length version of the protein weighing 280 kDa, and CHD8S (Duplin), a 110 kDa protein of the NH2-terminal chromodomain region. However, the latter does not have a counterpart in humans [46]. CHD8 interacts with CHD7 and FAM124B (family with sequence similarity 124B) as a complex [47,48]. However, the functional implication of this interaction in the context of inner ear development and deafness remains elusive.

Chd8 has been found in the proteome of HCs from P4–P7 organs of Corti derived from a *Pou4f3-eGFP* transgenic mouse model [49]. It has been suggested as a candidate deafness gene (DFNA53) with a possible functional role in HCs. In another study, RNA-seq data showed that CHD8 knock-out hES cell-derived neuroectodermal cells are enriched in genes for “inner ear morphogenesis” gene-ontology terms such as *PAX8*, *FGF9* and *MYO15A* [50]. This observed increase in “inner ear” genes might arise since CHD8 has been shown to interact directly with β-catenin and be recruited to the promoter regions of numerous β-catenin-responsive genes acting as a repressor of both β-catenin and the Wnt pathway [51,52]. Several identified genes (*PAX8, DLX6, FGF9*) are upstream or downstream effectors of β-catenin/Wnt pathway modulation [53,54].

In a recent study, the deletion of Chd8 in mice under the control of the Atoh1 promoter—expressed explicitly in HCs and spiral ganglion neurons—did not affect ABR [55]. Moreover, the development of HCs and auditory neurons seemed normal. Therefore, Chd8 does not appear to be necessary for cochlea development and hearing function in mice, while it seems essential in humans. The functional role of CHD8 in inner ear development and hearing loss remains to be established.

## 3. SWI/SNF Complexes

First discovered in yeast and further characterised in drosophila and mammals, switch/sucrose non-fermentable (SWI/SNF) are widely studied evolutionarily conserved protein complexes [56]. Three mammalian SWI/SNF complexes have been identified: BAF (BRG1/BRM-associated factor, also called Brahma-associated factor), PBAF (polybromo-associated BAF) and, very recently, non-canonical BAF (ncBAF or GBAF). BAF and PBAF co-exist in all cell types. They all contain a catalytic ATPase subunit—**S**WI/SNF-related, **M**atrix-associated, **A**ctin-dependent **R**egulator of **C**hromatin, subfamily **A**, member **2**—SMARCA2 (Brm) or SMARCA4 (Brg1)—that generates the energy for chromatin remodelling through the hydrolysis of ATP (Table 2). SWI/SNF complexes also contain several nucleosome-binding domains, including DNA-binding domains, such as AT-rich interactive domains (ARIDs), zinc-finger domains or high-mobility group box domains (HMGs). In addition, they have histone-binding domains, such as bromodomains (BRDs), plant homeodomains (PHDs) and chromodomains. SWI/SNF remodellers facilitate chromatin access as they slide and eject nucleosomes and are used for either gene activation or gene repression [57,58,59,60]. SWI/SNF complexes contain up to 15 subunits, and many of these are encoded by gene families and can be replaced by their paralogues. Therefore, hundreds of potential assemblies become a way to ensure complex specificity. Indeed, distinct subunit compositions occur at different time points during development and in various tissues. Many resulting complexes are unique to specific tissues or biological functions such as neural, heart or muscle development [61,62,63,64] or ES cell pluripotency [58].

Numerous SWI/SNF genes are mutated in a rare congenital developmental disorder, the Coffin–Siris syndrome (CSS), with variable degrees of hearing loss affecting up to 50% of patients [65,66]. With the advancement of next-generation sequencing genetic screening, mutations in other BAF/PBAF partner genes such as *SMARCC2* (BAF170), *SMARCG2* (DPF2) and *PHF6* have been found in patients affected with CSS-like syndrome affected with hearing loss with varying degree [67,68,69]. These clinical and genetic findings demonstrate that the SWI/SNF complex subunits are imperative in chromatin remodelling activity and developing inner ear components, leading to hearing loss.

In the inner ear, *in situ* hybridisation experiments show that Brg1 and Baf170 mRNAs are highly expressed in the developing otic vesicle starting at E8.5 in mice, and in the cochleovestibular ganglion at E10 [59,70]. Later, at E14.5–16.5, mRNAs remain high in the spiral ganglion. No mRNA could be detected in the organ of Corti at these stages. Postnatally, histological data show that Brg1 is highly expressed in HCs and SCs—both in the auditory and vestibular portions of the inner ear and spiral ganglion neurons at P4 [59,71].

Conditional Brg1 deletion in HCs using the Atoh1-cre line leads to profound deafness and rapid cochlear HC degeneration [59]. In addition, confocal and scanning electron microscopy images have shown that Brg1 inactivation results in numerous HC apical morphology defects, including disorganised OHC stereocilia, a flattened form instead of V-shaped stereocilia bundles at P3, and round-shaped hair bundles at P8, resulting in the abolishment of the cell-intrinsic polarity of OHCs [59]. Similarly, Brg1 deletion in the vestibule resulted in hair cell death and abnormal stereocilia in surviving HCs [71].

The functional contribution of the SWI/SNF complex for otic development was shown to be dependent on a physical interaction between Brg1 and Baf170 proteins with Six1 and Eya1, two transcription factors essential for otic neuronal differentiation. Eya1 and Six1 may act by recruiting and interacting with the SWI/SNF complex to induce the transcription of Neurogenin 1 and Neuro D1, two bHLH transcription factors regulating spiral ganglion neurogenesis [72]. Indeed, ectopic expression of Six1 and Eya1 in combination with Brg1 and Baf170 in nonsensory epithelial cells of mouse cochlear explant (GER cells) resulted in the production of a majority of Tuj1+/Neurofilament+ neurons. Moreover, Sox2 was found to act synergistically with the combination of Six1, Eya1, Brg1 and Baf170 to induce 100% of GER cells reprogramming into neurofilament positive cells. Altogether, this work showed that the functional specificity of Brg1-containing BAF complexes is regulated by the tissue-specific transcription factors *Eya1* and *Six1* to regulate the initial activation of *Sox2* expression specifically in the otic ectoderm to specify proneurosensory fate. Recently, it has been shown that conditional deletion of Brg1 at E8.5 in the mouse otocyst results in a loss of Sox2 expression and dysregulation of proneurosensory fate, leading to abnormal apoptosis within the otic ectoderm [70].

The evidence described above indicates a functional implication of Brg1 in mouse otic development. Nonetheless, integrative experimental analysis in a humanised model and/or hESC/hiPSC-derived otic organoid model for other component subunits of the SWI/SNF complex mutated in CSS or CSS-like disorders would potentially unravel a common molecular mechanism of hearing loss phenotypes.

**Table 2 cells-12-00532-t002:** Summary of *SMARC* genes present in chromatin remodelling complexes and disease-associated mutations.

Gene	Protein Alias	Complex	Disease Mutation	Refs
SMARCA2	BRM	BAF/PBAF/ncBAF	Coffin–Siris syndrome, Nicolaides–Baraitser syndrome	[73,74]
SMARCA4	BRG1	BAF/PBAF/ncBAF	Coffin–Siris syndrome	[74,75]
SMARCC1	BAF155	BAF/PBAF/ncBAF		[76]
SMARCC2	BAF170	BAF/PBAF/ncBAF	Coffin–Siris syndrome, Coffin–Siris syndrome-8	[67]
SMARCB1	BAF47	cBAF/PBAF	Coffin–Siris syndrome	[74,75]
SMARCE1	BAF57	cBAF/PBAF	Coffin–Siris syndrome	[73,74,75]
SMARCD1, 2 or 3	BAF60A, B and C	BAF/PBAF/ncBAF	Coffin–Siris syndrome-11	[77]
SMARCG2	DPF2	BAF/PBAF/ncBAF		[68]
SMARCA1	SNF2L	NURF/CERF	Coffin–Siris syndrome-like phenotype; Rett syndrome-like phenotype	[78]
SMARCA5	SNF2H	ACF, CHRAC, RSF, NoRC, WICH	Neurodevelopmental syndrome with mild facial dysmorphia	[79]

## 4. INO80/SWR Complexes

INO80 and SWR are multi-subunit complexes that remove and incorporate, respectively, the H2A.Z histone variants [80,81,82]. These variants affect nucleosome stability and are essential for early mammalian development [83]. Indeed, both INO80 and SWR are essential for stem cell renewal and pluripotency [84,85]. INO80 consists of the INO80 complex, while SRCAP and P400/TIP60 make up SWR (reviewed by [86]).

Compared with the other ATP-dependent chromatin remodellers, information regarding the involvement of INO80 in hearing and the inner ear is limited. One study found a potential role of INO80 variants in early-onset Ménière’s disease, an inner ear disorder affecting both the cochlea and the vestibule [87]. An earlier study investigating neurogenetic disorders reported microcephaly and developmental delay in patients with mutations in INO80 [88], although hearing loss was not reported. However, as with other mutations associated with microcephaly, such as *CDK5RAP2* [89], hearing loss might occur in only a subset of these patients. The loss of INO80 results in microcephaly due to unrepaired DNA breaks and increased apoptosis in neural progenitor cells undergoing symmetrical divisions. In contrast, asymmetric divisions—giving rise to a neural progenitor and a neuron—are unaffected [90]. While this loss of progenitor cells during development could explain microcephaly due to the loss of INO80, whether and how this affects the development of the inner ear and hearing function is not yet known.

The SRCAP complex is part of the SWR subfamily, and truncating mutations in SRCAP result in Floating–Harbor syndrome (FHS), in which some cases are associated with hearing loss [91,92,93]. SRCAP normally localises to the nucleus of cranial neural crest cells, while mislocalisation is observed in cells from FHS patients carrying SRCAP mutations [94]. The expression of cranial neural crest cell genes was additionally altered in these cells. Moreover, overexpression of an H2A.Z variant could rescue some of the defects observed in SRCAP mutants. A recent study found that SRCAP also localises to the centrosomes and midbody of cells during mitosis and that depletion of SRCAP in HeLa cells affected mitosis and cytokinesis [95]. In addition to its role in chromatin remodelling and stability, SRCAP could have additional roles in controlling cell division, which could help explain some of the observed symptoms in patients carrying *SRCAP* mutations. However, as with INO80, it is unclear if and how this could be linked to hearing loss.

## 5. ISWI Protein Complexes

The ISWI (imitation switch) protein was discovered in drosophila as the ATPase motor of the nucleosome and remodelling factor (NURF) complex [96]. In mammals, two homologues of ISWI comprise the closely related *SMARCA1* and *SMARCA5* genes that encode sucrose non-fermenting-like (SNF2L) and sucrose non-fermenting homologue (SNF2H), respectively (Table 2) [97]. There are two functionally essential domains in ISWI proteins: an N-terminal catalytic ATPase domain that mediates DNA interactions and a C-terminal HAND-SANT-SLIDE (HSS) domain, responsible for extranucleosomal DNA binding alongside H4 tail interactions [98].

Each of the two ATPase units SNF2L and SNF2H form heterodimers with either a BAZ-family transcription factor or a larger protein (BPTF, CECR2 or RSF1) capable of interacting with acetylated histones [99]. In mammals, SNF2H protein levels are copious and widespread, while SNF2L protein expression is more tissue-specialised and often less abundant [97]. Eight mammalian ISWI complexes have been identified thus far in humans and/or mice, in which SNF2L and SNF2H are present mutually exclusively to serve the function of the catalytic subunit [98].

Among these ISWI complexes, only CERF (Cecr2-containing remodelling factor complex) has been shown to play a role in inner ear development and neural tube closure [99,100]. CERF is composed of SNF2L and Cecr2. Cecr2 is detected from E8.5 in the entire otic vesicle and continues to be expressed in both the organ of Corti and the cochleovestibular ganglion. At E18.5, Cecr2 becomes restricted on the apical cochlear turn, where immature HCs are abundant, and gradually diminishes towards the basal turn, where most of the hair cells are mature, raising the potential involvement of Cerc2 and CERF complex activity for HC maturation and development [100]. Two distinct mutations in *Cerc2* have been shown to cause inner ear developmental defects with smaller and wider cochlea and stereocilia disorganisation in hair cells in the organ of Corti [100].

The fact that these defects are reminiscent of the misregulation of planar cell polarity (PCP) mutants suggests a possible influence of *Cecr2* on the PCP pathway. Surprisingly, microarray analysis of PCP genes shows that *Cecr2* exencephaly is unlikely to directly impact the PCP pathway [100]. However, as the study was conducted on an Affymetrix Mouse Genome 430 2.0 microarray, which only includes PCP-specific probes and this technique’s associated sensitivity, further validation with a more sensitive method such as RNA-Seq needs to be carried out to evaluate the effect of Cecr2 deficiency towards PCP pathway disruption.

## 6. Conclusions

It has been shown time and again that ATP-dependent chromatin remodellers are of paramount importance in developmental processes and their involvement in disease, and increasing evidence suggests their implication in the inner ear. Among others, new omics, biophysics and bioinformatics technologies are providing valuable tools to study epigenetic modulation via chromatin remodellers. However, this subject needs to be further investigated in the context of human disease—notably using organoid models, which are rapidly developing, to realistically recapitulate the developmental pathways involved in the emergence of a bona fide inner ear organ, although some challenges remain to be overcome, namely generating actual cochlear-identity hair cells. This would ultimately allow the high-throughput screening of molecules, gene and cell-based therapies, which would accelerate the drug discovery process for diseases involving chromatin remodeller defects.

## Figures and Tables

**Figure 2 cells-12-00532-f002:**
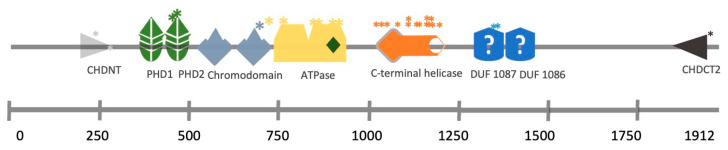
CHD4 structural domains and the location of SNPs identified in a cohort of SIHIWES patients. ***** Mutation position.

**Table 1 cells-12-00532-t001:** Main types of ATP-dependent chromatin remodellers.

	Specific	Function	Shared
**CHD**	Chromo	Binds methylated lysines in histone tails	ATPase * domain (DExx and HELICc )
**SWI/SNF**	Bromodomain and Helicase-SANT-associated (HSA) domain	Recognises acetylated lysines in histone tails	
**INO80**	HSA domain	Binds actin-related proteins	
**ISWI**	C-terminus HAND, SANT, SLIDE (HSS) domains	Recognises nucleosomes and internucleosomal DNA	

* The ATPase domain is constituted of two tandem RecA-like folds, a DExx box helicase and a HELICc (helicase superfamily c-terminal domain).

## Data Availability

Not applicable.

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
