# Peer review of "ATP-Dependent Chromatin Remodellers in Inner Ear Development"

_cells, 2023, doi:10.3390/cells12040532_

Round 1
Reviewer 1 Report
This is a well-written and through review of the family of proteins involved in ATP-dependent chromatin remodeling during inner ear development.
This review addresses all four main families of ATP-dependent chromatin remodelers, and beautifully summaries the current knowledge on how these families of proteins function, and pays particular attention to diseases related to the dysfunction of these proteins.
The figures are easy to understand and appropriate for the content.
Author Response
We would like to thank the reviewer for his enthusiastic comments.
Reviewer 2 Report
Interesting work.
About the atp dependet remodelling mechanism, this is involved not only in the development of inner but also in the noise injury. About this, you should mention this aspect mentioning this manuscript:
Pollarolo M, Immordino A, Immordino P, Sireci F, Lorusso F, Dispenza F. Noise-Induced Hearing Loss in Police Officers: Systematic Review. Iran J Otorhinolaryngol. 2022 Sep;34(124):211-218.
I thought the review article was comprehensive and well-written.The figures were designed and laid out well.
It was very easy to follow. I think the review article could easily be published as it is.
Author Response
Point 1:
About the atp dependet remodelling mechanism, this is involved not only in the development of inner but also in the noise injury. About this, you should mention this aspect mentioning this manuscript:
Pollarolo M, Immordino A, Immordino P, Sireci F, Lorusso F, Dispenza F. Noise-Induced Hearing Loss in Police Officers: Systematic Review. Iran J Otorhinolaryngol. 2022 Sep;34(124):211-218.
Response 1: We agree with the reviewer that chromatin remodelers are also involved in stressful situations, including acoustic trauma. We indeed discuss that for Chd7 and cite one reference that showed a specific vulnerability of hair cells and auditory neurons to stress in the absence f Chd7 (ref 39 in the review). The paper cited by the reviewer does not discuss or present the role of chromatin remodelers following acoustic trauma. Therefore, we did not include it in the review.